# Improving Skin Carotenoid Levels in Young Students through Brief Dietary Education Using the Veggie Meter

**DOI:** 10.3390/antiox11081570

**Published:** 2022-08-14

**Authors:** Akira Obana, Ryo Asaoka, Ayako Miura, Miho Nozue, Yuji Takayanagi, Mieko Nakamura

**Affiliations:** 1Department of Ophthalmology, Seirei Hamamatsu General Hospital, 2-12-12 Sumiyoshi, Naka-ku, Hamamatsu City 430-8558, Shizuoka, Japan; 2Department of Medical Spectroscopy, Institute for Medical Photonics Research, Preeminent Medical Photonics Education & Research Center, Hamamatsu University School of Medicine, 1-20-1 Handayama, Higashi-ku, Hamamatsu City 431-3192, Shizuoka, Japan; 3Faculty of Health Promotion Sciences, Department of Health and Nutritional Sciences, Tokoha University, 1230 Miyakoda-cho, Kita-ku, Hamamatsu City 431-2102, Shizuoka, Japan; 4Department of Community Health & Preventive Medicine, Hamamatsu University School of Medicine, 1-20-1 Handayama, Higashi-ku, Hamamatsu City 431-3192, Shizuoka, Japan

**Keywords:** skin carotenoid levels, elementary and junior high school students, Veggie Meter, dietary education, green and yellow vegetables, vegetable/tomato juice

## Abstract

The antioxidant and anti-inflammatory effects of carotenoid have been determined to aid in the prevention of a wide range of oxidative disorders, arteriosclerosis, obesity, and various types of cancers. In order to keep high carotenoid levels in the body, much of the vegetable and fruit (V/F) intake is mandatory. However, the actual intake of V/F is not enough in many countries. The aim of this study was to assess whether brief dietary education using the Veggie Meter (VM) that could measure skin carotenoid (SC) levels could induce the increase in carotenoid levels via V/F intake. Two hundred and sixty-one elementary and junior high school students (ages 7–14 years old) received brief educational session and SC evaluation by VM, and the changes in SC levels were examined after 6 months. The baseline VM scores ranged from 131 to 825, and the average significantly increased from 400.0 ± 124.7 (standard deviation) to 447.4 ± 140.4 at Month 6 (*p* < 0.0001). The percentage of increase at month 6 was negatively correlated with the baseline values (*r* = −0.36, *p* < 0.0001). This finding implies that subjects who became aware of their inferiority tended to make a significant effort to change their behavior. The multivariate logistic regression analysis demonstrated that subjects taking much of green and yellow vegetables, drinking vegetable/tomato juice, and eating any fruit had higher VM scores than the average value. In conclusion, the educational approach using VM was supposed to be an effective method of raising awareness of the V/F shortage and increasing V/F intake that could indue the increase in SC levels.

## 1. Introduction

Carotenoids are organic pigments produced by plants and algae and have a basic structure made up of 8 isoprene molecules and 40 carbon atoms. The number of double bonds can vary between 9 and 11, depending on the carotenoid species. The conjugated carbon chains of carotenoids quench singlet oxygen and other radical species [1,2,3]. Carotenoids have antioxidant and anti-inflammatory effects in the bodies of organisms [4,5]. Humans cannot synthesize carotenoids [6]; thus, they must obtain them from foods such as vegetables, fruits, eggs, and pink- or red-fleshed seafood. Humans consume about 30 carotenoids in their diet. The antioxidant and anti-inflammatory effects of carotenoid have been determined to aid in the prevention of a wide range of oxidative disorders, arteriosclerosis, obesity, and various types of cancers [7]. The human eye has three carotenoids, namely lutein, zeaxanthin, and meso-zeaxanthin, which work to maintain healthy vision and prevent age-related macular degeneration by absorbing blue light and exerting an antioxidant effect [8,9].

In order to provide sufficient carotenoids, the consumption of generous amounts of vegetables and fruits, major sources of carotenoids, is required. Much intake of vegetable and fruit (V/F) is effective to prevent hypertension, overweight and obesity in children [10,11], and lifelong V/F consumption is inversely associated with the incidence of chronic diseases, including diabetes, hypertension, coronary heart diseases and cancers [12]. Therefore, establishing healthy dietary habits in childhood is potentially important. Health organizations in many countries have set target values for vegetable and fruit intake (V/F intake). For instance, in Japan, the Ministry of Health, Labour and Welfare recommends at least 350 g of vegetables per day in its health promotion project titled “Health Japan 21 (the second term)” [13]. Nonetheless, the average Japanese vegetable intake was merely 280.5 g (288.3 g for males and 273.6 g for females, as per National Health and Nutrition Survey 2019), and this shortage was remarkable, particularly in younger generations aged 20–40. Many countries report vegetable and fruit shortages [14].

In this token, accurate estimation of the quantity of V/F intake is important as a first step. There are several methods to assess V/F intake, including diet record, diet recall, food frequency, diet history, and duplication methods, but each has its own advantages and drawbacks. However, a possible problem associated with this approach is that intimate assessments take time, labor, and cost, whereas simple assessments are deemed inaccurate. Measuring biomarkers is an alternative method of assessing V/F intake, and blood carotenoid concentration is one of the useful biomarkers for V/F intake [15,16,17,18,19]. Although plasma or serum carotenoid concentrations can be accurate measures, taking blood samples is mildly invasive and costly. In addition, it is undoubtedly difficult to apply the blood sampling approach to subjects like children and large-scale studies. On the other hand, the measurement of skin carotenoid (SC) levels is known to be a non-invasive rapid screening alternative that can be conducted using resonance Raman spectroscopy (RRS) or reflection spectroscopy (RS) methods [20,21,22,23,24,25]. There are various types of carotenoids in human skin, including lycopene; alpha-, beta-, gamma-, and delta-carotene; beta-cryptoxanthin; lutein; and zeaxanthin. Previous studies have shown that SC levels measured by RRS and RS are highly correlated with blood carotenoid concentrations, as measured by high-performance liquid chromatography (HPLC) [26]. Furthermore, a close association between RRS measurement and skin biopsy specimen was validated [27,28]. According to previous studies [23,29], SC levels reflect dietary intake in the previous 2–4 weeks. The drawbacks of RRS are the expensive cost of laser excitation and the highly sensitive detection schemes required to measure the Raman signals of carotenoid vibration. In contrast, a pressure-mediated RS (Veggie Meter, Longevity Link Corporation, Salt Lake City, Utah) is a relatively new device that can measure SC levels with less complexity and cost than an RRS. The measurements are displayed as a Veggie Meter (VM) score (arbitrary unit). It has been confirmed that the VM can be used in public health studies as well as studies with preschool and school children [30,31,32,33,34], where the validity was ensured through the associations between RS measures and dietary carotenoids, V/F intake as calculated from a validated food frequency questionnaire, and plasma carotenoids [35,36]. Thus, quantifying V/F intake using a VM can raise dietary awareness and may promote the efficiency of dietary education even in children. The aim of this study was to assess the effectiveness of brief dietary education using the VM for students aged 7–14 years in elementary and junior high school and to investigate the increase in carotenoid levels.

## 2. Subjects and Methods

### 2.1. Subjects

In this study, students in the second and fifth grades of elementary school and first and second grades of junior high school at the Faculty of Education at the Shizuoka University in Hamamatsu were surveyed, and 261 out of 353 students were included (129 boys and 132 girls). The subjects in the second elementary, fifth elementary, first junior high, and second junior high grades were 7 or 8 years old, 10 or 11 years old, 12 or 13 years old, and 13 or 14 years old, respectively.

### 2.2. Methods

Just prior to the start of this study, homeroom teachers explained the abstract of this study to students, including the outlines of scientific research work, the objectives and significance of this study, and the actual methodology, and they delivered informed consent forms for parents and assent forms for children to all students. A dietary survey form was also distributed. The students handed them over to their parents at home. Students were only enrolled in this study if their parents agreed to participate and returned signed consent forms to their classroom teachers.

Before the SC levels were measured on the first day of the examination, a registered dietitian talked briefly (approximately 10 min) to all students in elementary school classrooms and via intranet monitor in junior high school about the importance of V/F intake for health and the method to measure SC levels. During this dietary education, the registered dietitian explained the following topics. (1) Vegetables and fruits are rich in many nutrients that are beneficial to one’s health. (2) This study examines the state of students’ V/F intake and makes subjects aware of their situation. (3) The V/F intake state was assessed 3 and 6 months after the baseline. (4) Subjects with low SC levels, indicating a shortage of V/F intake, were advised to take more V/F. (5) SC level, even if low, does not indicate poor health, and numerical does not indicate any inferiority or superiority of personal ability. (6) Subjects can withdraw from this study at any time.

All students were given a report paper with their VM score and rank in their grade (Figure 1a), as well as a flyer explaining the importance of a balanced diet with a 350 g vegetable figure (Figure 1b).

All students’ VM scores were ranked into five levels for easy understanding of their current status when compared to others’ scores. In addition, a short comment was included with each rank in the paper to encourage subjects. The ranking was determined using our previous data on 985 subjects [37], and the subjects were then classified into five groups of every 20% distribution. The highest 20% of subjects (rank A) had a veggie score greater than 507, the second highest group (B) had a score between 396 and 506, the third group (C) had a score between 311 and 395, the fourth group (D) had a score between 240 and 310, and the lowest group (E) had a score less than 239. Students were instructed to show their parents these reports, which can help parents understand their child’s state.

SC levels were measured three times: at the baseline, 3 months later, and 6 months later. The date of examinations as well as the weather of the day is shown in Table 1. A dietary survey was conducted at the first and third examinations.

This study was conducted according to the guidelines of the Declaration of Helsinki and approved by the Institutional Review Boards of Seirei Hamamatsu General Hospital (No. 3601) and Tokoha University (No. 2021-002H).

### 2.3. Measurement of Skin Carotenoid Levels

The SC levels were measured using the VM according to the manufacturer’s instructions. The principles of this device have been described elsewhere [23]. Briefly, VM detects dermal carotenoids that are highly correlated with serum carotenoid concentrations measured by HPLC [26]. Serum or plasma carotenoid concentrations are well-established as biomarkers of V/F intake [15,16,17,18,19]. Therefore, the SC levels measured by VM can well serve as a biomarker for V/F intake [35,36].

All measurements were taken using the same VM. Before measuring each class, calibration was performed with the provided dark and white reference materials. There were two classes in one grade of elementary school and three classes in one grade of junior high school. All subjects washed their hands with soap and disinfected their fingers with disinfectant. The researcher then checked the subjects’ hands and found no contaminations. The subjects then inserted their finger into the device’s finger cradle and had the tip pushed against the convex contact lens surface with the help of a spring-loaded lid. The modest pressure applied to the fingertip reduced the blood perfusion of the measured tissue volume, preventing the strongly absorbing blood from interfering with the measurement of SC levels. The middle finger of the left hand was measured in the fifth grade of elementary school and both grades of junior high school, while the thumb of the left hand was measured in the second grade of elementary school. The VM score was calculated as the average of three consecutive measurements. Examinations were conducted in a dimly lit room away from the window to avoid direct sunlight. Measurements were completed in a classroom setting with minimal disruption to regular instruction. The percentage of increase in VM scores at month 6 was calculated as follows: (VM score at Month 6 − VM score at baseline)/VM score at baseline.

### 2.4. Dietary and Life Habitual Survey

Parents completed the survey form based on their own considerations while also discussing with their children as deemed appropriate. The contents of the form are shown in Table 2. As shown in the table, several answer choices were adapted.

### 2.5. Statistical Analyses

The VM scores were compared between two categories using the unpaired Student’s *t*-test and between three and more habitual categories using one-way analysis of variance (ANOVA), with multiple comparisons adjusted using Bonferroni’s method. Changes in VM scores with time were tested by one-way repeated ANOVA with multiple comparisons adjusted using the Bonferroni’s method after the logarithmic transformation of VM scores. The correlations between two numerical categories were performed using Pearson’s correlation test. A multivariate logistic regression analysis was performed, with the dependent variable “whether VM score was above the baseline average (400) or not” and the 15 independent variables: grade, body mass index (BMI) percentile, vegetable preferences (like, dislike, or neither), the number of vegetable cups taken (five cups or more a day or fewer than that), the number of green and yellow vegetable cups taking (three cups or more a day or fewer than that), taking any fruit in the previous 2 weeks (yes or no), taking *Citrus unshiu* in the previous 2 weeks (yes or no), the frequency of drinking vegetable/tomato juice (three times and more a week or less than that), the frequency of drinking 100% orange juice (three times or more a week or less than that), the frequency of drinking fruit juice other than orange (three times or more a week or less than that), the frequency of eating outdoors (four times or more a week or less than that), the frequency of eating take-out meals (four times or more a week or less than that), feeling of any stress (yes or no), improving the child’s dietary habits by parents (yes or no), and smoking by family members (yes or no). Subsequently, the optimal model to describe whether the VM score was above average (400) or not was selected from all possible 2^15^ combinations of the variables (i.e., round-robin method) based on the second-order bias-corrected Akaike Information Criterion (AICc, the corrected form of the AIC) index. The AICc was used because it provides an accurate estimation even when the sample size is small [38]. Subjects who did not respond to any of the 15 questions were excluded from the analyses. Statistical analyses were performed using the IBM Statistical Package for the Social Sciences software, version 25, and the R software, version 4.1.3, and a *p*-value of less than 0.05 was considered statistically significant.

## 3. Results

### 3.1. Baseline Characteristics of the Included Subjects

Eleven subjects received two measurements, and one subject received only one measurement because they were absent from school on the examination days. In 249 subjects, three consecutive measurements were achieved, and further analyses were performed on these subjects. The baseline characteristics of the 249 subjects are shown in Table 3.

### 3.2. Baseline VM Scores

The baseline VM scores are shown in Figure 2 and Table 4. The VM scores ranged from 131 to 825 (median, 386), with an average of 400.0 and a standard deviation of 124.7. VM scores distributed with a slight skew to higher levels, which is consistent with our previous findings in adults [37]. No statistically significant difference was noted in the VM scores between male and female (*p* = 0.557, *t*-test). The VM scores of the second grade of elementary school students were significantly higher than those of the fifth grade of elementary school students and the second grade of junior high school students and tended to be higher than those of the first grade of junior high school students (*p* = 0.005; one-way ANOVA, *p* = 0.006, 0.039, and 0.068, respectively; Bonferroni’s test). There was no significant correlation between VM scores and percentile of BMI (*r* = −0.004, *p* = 0.955).

### 3.3. The Change in VM Scores from Baseline to Month 6

At month 6, VM scores were noted to increase by more than 10% from the baseline value in 136 subjects (55%) and decreased by more than 10% in 44 subjects (28%). The change in VM scores between baseline and month 6 was within 10% in 69 subjects (28%). The change in VM scores is shown in Table 5. In addition, VM scores increased with time and were significantly higher at months 3 and 6 than at baseline (*p* < 0.0001, one-way repeated ANOVA; all *p* < 0.0001, Bonferroni’s test).

Figure 3 depicts the ranking distribution at three different time points. Subjects with Rank A increased with time, while subjects with Rank C and D decreased. The percentage of increase in VM scores at month 6 had a significantly negative correlation with the baseline values (*r* = −0.36, *p* < 0.0001, Figure 4). Figure 5 depicts the percentage of increase based on five ranking levels. The percentage of increase in Rank D and C subjects was significantly higher than that of Rank A subjects (*p* < 0.0001, one-way ANOVA; *p* = 0.001 and <0.0001, respectively, Bonferroni’s test).

### 3.4. The Correlation between VM Scores and Dietary and Life Habitual Surveys

The results of dietary and habitual surveys are shown in Table 2. The VM scores for each question were statistically analyzed, and *p*-values are shown in Table 2.

Table 6 demonstrates the significant variables in the multivariate logistic regression models with model selection of factors potentially associated with VM scores greater than the average baseline value. The model identified three factors that had significant positive associations with high VM scores, that is, drinking vegetable/tomato juice three times or more a week, eating fruit in the previous 2 weeks, and taking three cups or more of green and yellow vegetables daily, while the grade was correlated negatively with high VM scores. Furthermore, eating *Citrus unshiu* in the previous 2 weeks and improvement of the child’s dietary habits by parents showed no significant association, and the other nine factors were not included in the analysis.

## 4. Discussion

In this study, we have assessed the effectiveness of brief dietary education using the VM to increase SC levels. In this educational program, we provided students and their parents with brief information on the importance of V/F intake for health promotion as well as their VM scores, which represent their current carotenoid intake. As a result, in the repeated measures, VM scores showed a significantly increasing trend, which only indicates the usefulness of this educational approach in terms of improving dietary habits. Furthermore, we found that eating green and yellow vegetables and fruits and drinking tomato/vegetable juice were associated with greater VM cores than the average. This result seems reasonable because green and yellow vegetables and fruits are the primary sources of carotenoids in the human diet, and tomato/vegetable juice was thought to be a useful carotenoid supplier.

Persistent behavior change is challenging in general, especially when it comes to dietary habits. Personal counseling, e-mail or telephone counseling, seminars, group discussions, posters, flyers, menu changes in cafeterias, and various strategies in grocery stores have all been proposed to improve V/F intake [39], and studies using these methods have shown an increase in V/F intake [40,41,42,43,44]. Among these, personal counseling based on dietary assessment was suggested to be an effective method, but it is obviously time-, labor-, and cost-consuming. Although group seminars can provide a large population with knowledge of a healthy diet at one time, their effectiveness is limited. Recently, an online personal education program using a smartphone has been tried in the workplace and school [31], but the use of such an education program has not been widely adopted. In contrast, VM can easily quantify V/F intake, allowing subjects to understand their current state of V/F intake as the VM score. In addition, the comparison with other subjects provided a clear awareness of the shortage of V/F intake to subjects with low scores. The percentage of increase in VM scores at month 6 was found to be inversely correlated with the baseline VM scores. This result implies that subjects who become aware of their inferiority tended to make significant efforts to change their behavior. Thus, quantifying V/F intake using the VM scores is an effective approach to motivate V/F intake.

There have been several studies that used SC level measurement in dietary education in children. For one, Bakiri-T et al. [31] reported an increase in vegetable consumption as a result of the VM intervention, but no VM score details were provided. Jones et al. [34] reported an increase in VM scores between the baseline and second measurements but did not show a continuous increase at the third measurement. The other two studies failed to show an increase in SC levels and V/F intake [30,45]. We could show a gradual increase in VM scores in three consecutive measurements in children using the VM intervention. According to these previous studies, we considered some possible explanations for the present successful increase in VM scores. First, the goal of this project was to motivate not only students but also parents to take a lot of V/F. All students in this present study lived with their parents and ate meals prepared by their guardians. The elementary school students ate school lunch, and junior high school students ate “bento” for lunch on Monday to Friday. It was speculated that the letter to parents changed their perceptions, resulting in increased V/F intake. Indeed, a previous study by Bakirci-Taylor et al. [31] showed that parents’ awareness can have a significant effect on their children’s V/F intake (3–8 years old). On the other hand, Hopkins [30] failed to increase SC indices in high-school students, but the relatively young age of the present subjects was consistent with the previous study by Bakirci-Taylor et al. This may be because students in higher grades, such as high school and university, have more chances to eat out and tend to eat with their own preference than elementary and junior high school students. It was assumed that the current subjects would follow their parents’ advice. Second, body shape uniformity was considered another factor. The number of obese subjects was small. In adults, VM scores were inversely correlated with BMI [37,46], but no significant correlation was shown between VM scores and BMI percentile in children. Dietary changes could easily increase VM scores in the present subjects without obesity. Third, the present subjects were students from elementary and junior high schools of the Faculty of Education, National University Corporation, Shizuoka University. Generally, legal residents in Japan can enter public elementary and junior high schools without any restriction, but the present subjects passed the entrance examination to this reputable school and were judged to have higher intellectual levels than the average. Most of the subjects belong to relatively wealthy families, and their parents are thought to be concerned about their children’s education and health. Therefore, students and parents well understood the value of vegetables and changed their behavior.

The multivariate logistic analysis revealed that odds ratios for achieving VM scores higher than the average (400) were 7.30 when drinking vegetable/tomato juice three times or more a week, 3.54 when taking any fruits, and 2.89 when taking three cups or more of green and yellow vegetables per day. The investigation into the correlation between V/F intake and VM scores in children has not fully been proven. Martinelli et al. [32] showed a significant positive relationship between vegetable intake and VM scores, but not between green vegetable and fruit intake and VM scores. May et al. [33] showed no correlation between vegetable intake and VM scores. In contrast, Pitts et al. [35] showed a significant correlation between VM scores and carotenoids and V/F intake estimated by food frequency questionnaire. Rush et al. [36] found positive correlations between VM scores and weekly intakes of deep green leafy vegetables (*r* = 0.242) and carrots and pumpkin (*r* = 0.202). The importance of eating green and yellow vegetables and fruits was also confirmed in the present results. In contrast, taking five cups or more of vegetables a day was not a significant factor associated with high VM scores. In this question, vegetables included all types of vegetables other than green and yellow ones. Since light-colored and root vegetables contain fewer carotenoids, green and yellow vegetables are considered more important for carotenoid consumption. We also asked about *Citrus unshiu* consumption because Hamamatsu city is known for its high satsuma mandarin production, and Hamamatsu residents have a habit of eating a lot of *Citrus unshiu*, which contains a high amount of β-cryptoxanthin [47]. However, no significant association was found between *Citrus unshiu* taking and VM scores. According to another study, vegetable/tomato juice was considered a useful alternative to eating green and yellow vegetables [26].

The multivariate logistic analysis demonstrated that high VM scores were negatively associated with grade. However, there were no significant differences between fifth elementary, first junior high, and second junior high school students, as shown in Table 4, although VM score of second grade elementary school students (7, 8 years old) was significantly higher than that of fifth elementary (*p* = 0.006) and second junior high school students (*p* = 0.039). May et al. [33] reported that preschool children (mean age: 4.1 years old) showed higher VM scores than middle-school (mean age: 12 years old) and high-school students (mean age: 15 years old). Preschool and low-grade elementary school students may have had higher VM scores than older ones, but further research is warranted in a population with a wider age range to draw a conclusion regarding the effect of age on VM scores.

Sex could influence VM scores. May et al. [33] showed that men had significantly higher VM scores than women in preschool children. However, Martinelli et al. [32] showed no sex differences, and sex was not a significant factor in this present study. In our previous studies in adults [37,46], women had higher VM scores than men had. Further research is needed to shed light on the effect of sex on VM score. Previous studies showed no significant difference in VM scores among races [32,33]. May et al. [33] reported that the mean VM scores for preschool, middle-, and high-school Black and White students were 266, 219, and 216, respectively. Martinelli et al. [32] reported a mean VM score of 210 in 9–12 years old White, Hispanic, and other students. The present VM scores in Japanese children appeared to be higher than in other previous studies. However, because the values were obtained in different studies using different VMs, a direct comparison could not be made.

This study has several limitations. First, this is a single-arm study with no controls. Second, we made a judgment based on a five-level ranking and conveyed it to the subjects so that students and their parents could easily understand the present status and compare it to others. However, because the ranking was derived from the data on adults, this judgment should be investigated further in the future. Despite this reservation, we believed that the current judgment was useful in motivating subjects to take V/F because subjects with lower ranks had higher increasing rates of VM scores at month 6. Third, the social backgrounds of the present subjects were thought to be distinct from those of the general population. They belonged to relatively wealthy families, and their parents may have been concerned about their children’s education and health. Therefore, we intend to evaluate the present program in public schools. Fourth, the ring finger of the non-dominant hand has recently been recommended as a standardized method [48]. However, this study was planned prior to standardization, and we measured the middle finger of the left hand. There has been no consensus about the difference in VM scores in different fingers [49], but our preliminary study found no difference among the eight fingers of both hands (except little fingers, unpublished data). Fifth, VM scores were verified in the US population by dietary assessment [35], but not in the Japanese population. The present program ended in month 6. It is well known that persistent behavior change is difficult to achieve. We intend to investigate VM scores in the same subjects after a 1-year interval to assess the present program’s long-term impact.

## 5. Conclusions

In conclusion, this study showed a gradual increase in SC levels in children using the VM intervention. The present educational approach involving awareness of V/F shortage by quantification using the VM is an effective method to change V/F taking behavior and increase carotenoid levels. This approach saves time and labor and is thus applicable to a large cohort, including children. VM scores for taking three cups or more of green and yellow vegetables a day, eating fruits, and drinking vegetable/tomato juice of three times or more a week were higher than the average. The effect of age on VM scores needs further investigation in a population with a larger age range.

## Figures and Tables

**Figure 1 antioxidants-11-01570-f001:**
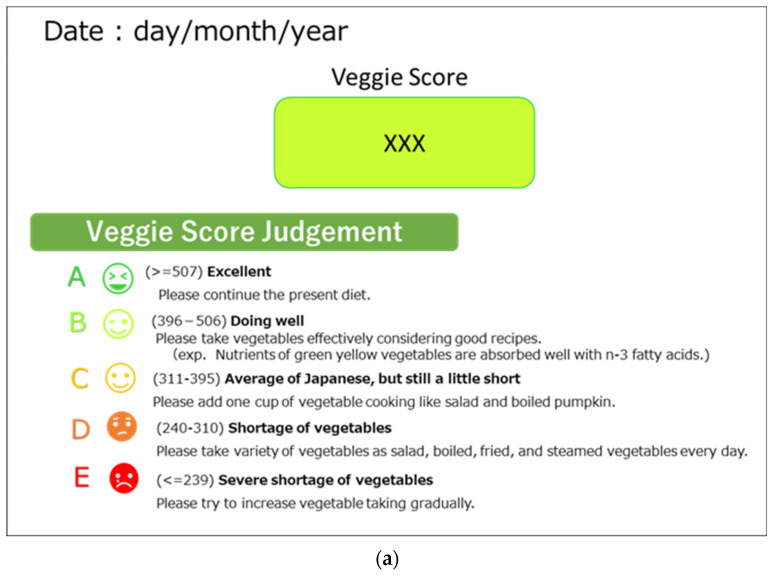
(**a**) A report used to inform the results. (**b**). A flyer describing the amount of vegetables and the importance of balanced diet. The original paper was written in Japanese.

**Figure 2 antioxidants-11-01570-f002:**
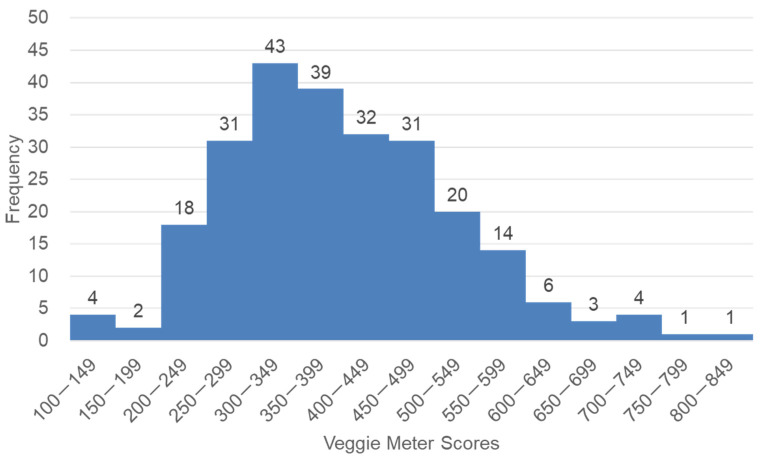
A histogram of the Veggie Meter scores of 249 students. The scores show a slight skew toward higher levels.

**Figure 3 antioxidants-11-01570-f003:**
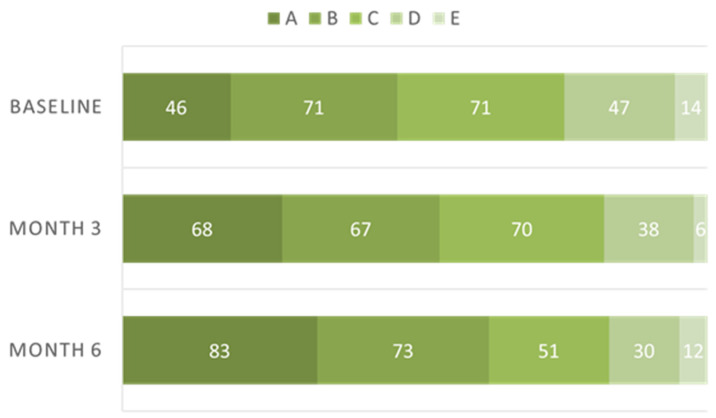
The ranking distribution of Veggie Meter score at three time points. The specifics of each rank were depicted in Figure 1a.

**Figure 4 antioxidants-11-01570-f004:**
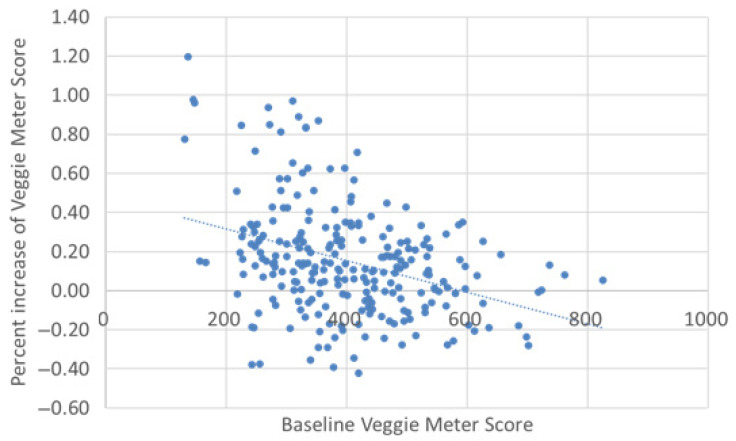
The correlation between baseline Veggie Meter score and percentage of increase in the scores. The percentage of increase had a significantly negative correlation with the baseline values (*r* = −0.36, *p* < 0.0001).

**Figure 5 antioxidants-11-01570-f005:**
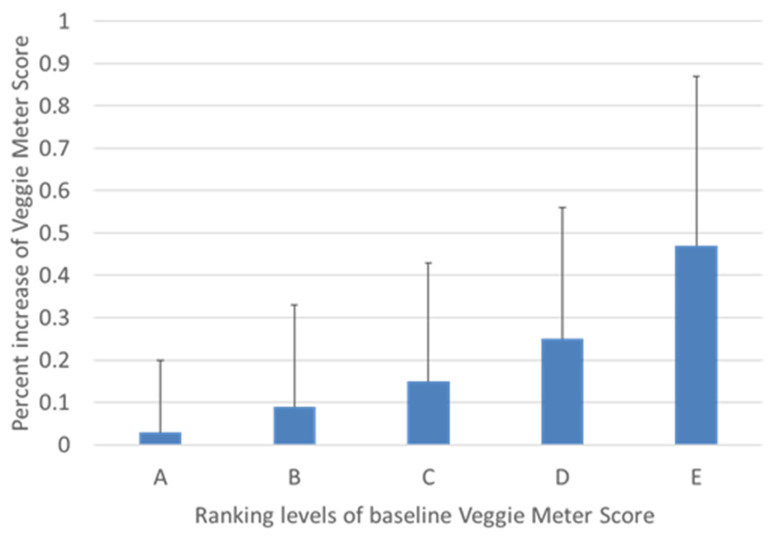
The percentage of increase in Veggie Meter score based on five baseline rankings. The percentage of increase in subjects with Rank D and C was significantly higher than subjects with Rank A (*p* < 0.0001, one-way ANOVA, *p* = 0.001 and <0.0001, Bonferroni’s test).

**Table 1 antioxidants-11-01570-t001:** Examination dates and conditions.

Elementary School	Baseline Examination	Second Examination	Third Examination
Date	1 June 2021	15 July 2021	30 November 2021
Weather	Clear	Clear	Clear
Temperature (°C)	26.2	26.8	21.2
Humidity (%)	44.2	53.8	42.5
**Junior High School**			
Date	11 May 2021	6 July 2021	10 November 2021
Weather	Clear	Clear	Clear
Temperature (°C)	23.5	29.0	18.4
Humidity (%)	42.5	65.8	39.8

Conditions were measured in the examination room.

**Table 2 antioxidants-11-01570-t002:** Dietary and life habitual survey and Veggie Meter scores.

	Question	Choices	Results at the Baseline		*p*-Value (Significantly Different Pair in Post Hoc)	Results at 6 Months Later		*p*-Value (Significantly Different Pair in Post Hoc)
			No. of Subjects	Veggie Meter Score Min–Max Median Mean ± Standard Deviation		No. of Subjects	Veggie Meter Score Min–Max Median Mean ± Standard Deviation	
Q1	Does your child like vegetables?	1, No	30	147–761 360 381.0 ± 155.5	0.262	27	160–777 321 371.5 ± 155.7	0.001 (1–2, 1–3)
		2, Neither yes nor no	78	217–736 381 389.6 ± 111.1		81	151–833 427 441.1 ± 139.5	
		3, Yes	138	131–825 403 409.6 ± 125.1		139	189–870 462 465.9 ± 133.8	
Q2.1	How many vegetables did your child eat daily in these 2 weeks? Answer the number of cups. One cup contains 70 g of vegetables.	1, <1	3	223–398 331 317.4 ± 88.3	0.014 (2–3)	9	160–440 288 273.9 ± 86.9	<0.0001 (1–2, 1–3, 1–4, 1–5)
		2, 1–2	67	136–698 335 360.3 ± 116.4		77	197–821 407 421.9 ± 134.1	
		3, 3–4	89	217–825 406 416.0 ± 127.1		113	151–870 455 459.4 ± 139.0	
		4, 5–6	45	131–723 437 425.9 ± 130.6		37	190–777 431 473.6 ± 126.3	
		5, ≥ 7	45	166–701 397 407.2 ± 116.2		9	375–798 505 561.4 ± 142.3	
Q2.2	How many green and yellow vegetables did your child eat daily in these 2 weeks? Answer the number of cups. One cup contains 70 g of vegetables.	1, <1	6	260–391 321 319.1 ± 43.9	<0.0001 (2–3, 2–4, 2–5, 3–5)	10	160–440 290 286.8 ± 77.5	<0.0001 (1–3, 1–4, 1–5, 2–4, 2–5)
		2, 1–2	26	136–698 303 312.9 ± 117.4		30	180–545 338 365.3 ± 100.0	
		3, 3–4	72	145–602 372 377.4 ± 93.9		89	197–821 421 438.4 ± 127.2	
		4, 5–6	72	131–761 398 419.6 ± 138.8		89	151–870 462 477.9 ± 146.9	
		5, ≥ 7	72	218–825 429 443.5 ± 121.4		29	190–798 523 522.3 ± 131.3	
Q3.1	Did your child eat any fruit in these 2 weeks?	1, No	34	147–761 330 341.9 ± 112.1	0.003	21	197–675 361 376.8 ± 123.7	0.011
		2, Yes	215	131–825 398 409.2 ± 124.4		223	151–870 437 454.7 ± 140.6	
Q3.2	Did your child eat *Citrus unshiu* in these 2 weeks?	1, No	158	131–825 371 383.5 ± 120.0	0.008	91	159–821 413 421.2 ± 137.5	0.022
		2, Yes	91	145–723 425 428.7 ± 128.2		157	151–870 451 462.2 ± 140.7	
Q4	How often did your child drink the following beverages in the last 2 weeks?							
	Vegetable/tomato juice	1, Almost none	181	131–825 370 380.2 ± 116.0	<0.0001 (1–3, 1–4)	165	180–870 417 423.1 ± 130.5	<0.0001 (1–3, 1–4, 2–4)
		2, Once a week	35	242–723 426 431.6 ± 126.1		30	160–833 403 441.1 ± 146.8	
		3, 3–4 times a week	15	295–685 538 495.6 ± 129.4		29	300–821 486 502.8 ± 129.4	
		4, Daily	10	325–761 533 528.3 ± 130.4		20	151–783 589 570.2 ± 154.5	
	Green juice	1, Almost none	234	131–825 387 403.0 ± 124.8	0.232	228	151–870 431 447.1 ± 138.1	0.035 (1–3)
		2, Once a week	4	156–515 356 345.8 ± 164.1		5	247–712 370 445.3 ± 197.8	
		3, 3–4 times a week	1	260		4	197–396 281 288.6 ± 92.7	
		4, Daily	0	-		0	-	
	Orange juice	1, Almost none	165	136–825 388 405.9 ± 127.0	0.561	174	151–870 438 446.4 ± 143.1	0.980
		2, Once a week	57	145–685 380 391.8 ± 120.0		52	242–783 408 441.3 ± 134.5	
		3, 3–4 times a week	17	131–597 342 372.7 ± 120.9		9	200–713 407 450.8 ± 154.1	
		4, Daily	2	391–523 457 457.0		3	313–563 336 404.1 ± 138.3	
	Other fruit juice	1, Almost none	147	131–736 382 390.7 ± 120.2	0.019 (1–4, 2–4)	145	151–833 428 438.2 ± 140.1	0.178
		2, Once a week	65	156–718 393 398.5 ± 120.3		59	200–777 426 436.3 ± 132.3	
		3, 3–4 times a week	22	282–698 375 419.6 ± 124.6		26	313–727 419 479.4 ± 135.4	
		4, Daily	9	342–825 523 545.9 ± 175.3		11	266–870 541 526.4 ± 192.6	
Q5	Did your child take supplements containing lutein?	1, No	232	131–825 385 398.9 ± 124.1	0.328	235	151–870 428 446.1 ± 139.3	0.257
		2, Occasionally	8	261–586 386 398.9 ± 108.1		5	280–538 327 398.5 ± 128.1	
		3, Sometimes	3	340–565 468 457.4 ± 112.9		4	288–821 518 536.1 ± 218.4	
		4, Daily	5	246–761 393 440.0 ± 203.5		2	603, 626 614.5	
Q6	Does your child eat breakfast?	1, Seldom	0	-	0.004 (2–3, 2–4)	0	-	0.028 (2–4)
		2, Less than half of the week	3	131–490 147 255.8 ± 202.6		6	151–610 307 331.2 ± 150.4	
		3, More than half of the week	11	166–761 342 383.0 ± 176.1		7	232–821 327 422.4 ± 212.9	
		4, Everyday	234	136–825 390 402.6 ± 120.8		234	160–87- 434 451.2 ± 137.3	
Q7.1	How often does your child eat out?	1, More than twice a day	0	-	0.096	1	389	0.947
		2, Once a day	1	248		1	484	
		3, 4–6 times a week	1	760		0	-	
		4, 1–3 times a week	76	156–718 381 394.3 ± 111.7		80	151–821 426 448.1 ± 158.8	
		5, Less than once a week	170	131–825 388 401.8 ± 127.8		162	190–870 428 445.5 ± 132.6	
Q7.2	How often does your child eat take-out meals?	1, More than twice a day	0	-	0.194	0	-	0.133
		2, Once a day	0	-		0	-	
		3, 4–6 times a week	1	459		1	820	
		4, 1–3 times a week	111	131–761 373 388.2 ± 131.1		115	160–833 428 445.3 ± 140.4	
		5, Less than once a week	137	166–825 386 409.2 ± 119.3		131	151–870 428 446.5 ± 138.0	
Q8	Does your child show appearance or feeling of any stress?	1, Much	9	166–825 490 455.1 ± 205.4	0.233	16	151–712 471 438.3 ± 165.4	0.755
		2, Some	124	145–761 386 397.5 ± 122.6		117	190–870 420 443.8 ± 145.6	
		3, Little	76	131–723 366 384.6 ± 121.7		74	160–833 430 450.2 ± 126.4	
		4, No	40	261–736 412 424.8 ± 112.3		38	230–777 466 464.2 ± 146.6	
Q9.1	How often does your child exercise?	1, Less than follows	26	218–536 328 358.3 ± 99.5	0.288	27	215–610 411 401.9 ± 114.1	0.444
		2, Once a week	66	145–736 383 396.4 ± 127.9		57	151–833 437 449.9 ± 155.5	
		3, 3–4 times a week	100	131–761 383 402.2 ± 129.3		98	160–821 448 458.6 ± 149.6	
		4, Every day	26	225–701 402 411.0 ± 126.3		27	307–646 453 463.1 ± 106.9	
		5, Playing competitive sport	22	228–825 441 440.9 ± 129.0		39	247–870 411 440.6 ± 128.3	
Q9.2	Is your child’s exercise time more than 1 h?	1, No	47	156–736 380 398.7 ± 134.7	0.283	53	225–833 413 451.3 ± 152.0	0.700
		2, Yes	105	136–825 406 417.6 ± 121.1		107	151–821 461 457.1 ± 133.9	
Q10	How many hours does your child spend outside?	1, Less than 30 min	8	156–530 340 358.7 ± 129.0	0.017	15	180–484 307 337.9 ± 103.4	0.001
		2, Less than 2 h	119	131–825 363 376.4 ± 119.0		234	151–870 442 454.4 ± 139.7	
		3, Less than 3 h	71	217–736 407 419.1 ± 124.8		0	-	
		4, Less than 4 h	40	248–723 431 435.7 ± 114.8		0	-	
		5, Equal or more than 4 h	11	256–761 386 433.3 ± 172.7		0	-	
Q11	What do you think about the dietary habits of your child?	1, I am not interested in it.	4	320–718 413 465.8 ± 175.3	0.013 (3–7)	0	-	0.064
		2, I have some interest but am not going to improve it.	32	145–736 336 364.6 ± 122.9		27	200–833 445 448.8 ± 160.2	
		3, I will improve it within 6 months.	72	131–761 343 368.0 ± 117.7		79	160–783 397 434.1 ± 147.6	
		4, I will improve it within 1 month.	28	242–698 434 433.1 ± 130.3		29	151–663 411 414.4 ± 117.3	
		5, I have improved for less than 6 months.	33	166–825 412 419.3 ± 131.7		47	190–661 451 444.4 ± 123.9	
		6, I have improved for more than 6 months.	40	226–656 392 399.5 ± 106.6		38	180–727 434 448.5 ± 127.2	
		7, I don’t feel any necessity to improve since I have already improved it.	38	218–723 431 442.7 ± 128.6		27	215–870 486 528.8 ± 149.2	
Q12	Does anyone in your family smoke?	1, Yes	24	166–57 403 372.5 ± 110.7	0.276	24	190–711 391 398.4 ± 139.5	0.043
		2, No	224	131–825 384 402.8 ± 126.3		223	151–870 431 453.3 ± 140.0	

**Table 3 antioxidants-11-01570-t003:** Baseline characteristics of the included subjects.

Grade	Number of Subjects	Age (Years)	Sex, Number of Subjects	Height Min–Max, Median, Mean ± SD (cm)	Body Weight Min–Max, Median, Mean ± SD (kg)	Body Mass Index Max–Min, Mean ± SD	Percentile of Body Mass Index Min–Max, Median
2nd, elementary	60	7, 8	Male, 28	112.0–130.0 122.0 121.8 ± 4.2	19.0–30.0 23.0 23.3 ± 2.8	12.8–20.1 15.4 15.7 ± 1.7	0.7–96.3 43.7 46.7 ± 29.5
			Female, 32	116.0–137.0 122.5 123.0 ± 4.5	19.8–29.5 23.0 23.1 ± 2.3	13.1–18.7 15.1 15.3 ± 1.3	3.6–90.8 39.2 41.3 ± 23.5
5th, elementary	53	10, 11	Male, 27	128.0–150.0 139.0 138.8 ± 5.2	24.0–48.1 31.0 32.7 ± 5.6	13.4–24.9 16.1 17.0 ± 2.7	0.8–96.2 38.1 44.8 ± 29.5
			Female, 26	127.0–152.0 140.0 139.8 ± 6.2	26.0–40.0 29.6 31.1 ± 4.2	13.7–19.9 15.9 15.9 ± 1.5	3.2–85.7 32.2 32.0 ± 21.5
1st, junior high	56	12, 13	Male, 30	137.0–175.0 155.0 155.0 ± 8.8	28.0–62.0 42.3 43.6 ± 8.8	13.7–25.8 17.4 18.0 ± 2.7	0.23–96.3 35.9 40.9 ± 35.9
			Female, 26	136.0–162.7 151.0 150.9 ± 6.1	28.0–68.0 41.4 43.4 ± 8.3	15.1–28.7 18.7 18.9 ± 2.9	5.5–99.1 53.0 50.2 ± 26.0
2nd, junior high	80	13, 14	Male, 39	137.0–174.0 160.0 160.2 ± 8.0	31.0–60.0 48.0 47.4 ± 7.4	14.2–22.4 18.8 18.4 ± 2.1	0.27–85.4 46.4 42.1 ± 27.4
			Female, 41	143.0–169.0 155.0 155.1 ± 5.2	33.0–56.0 44.3 45.1 ± 5.5	14.9–24.9 18.6 18.8 ± 2.3	1.4–94.3 40.2 41.7 ± 26.5

**Table 4 antioxidants-11-01570-t004:** Baseline Veggie Meter scores of subjects who achieved three consecutive measurements.

	Number of Subjects	Min	Max	Median	Mean	Standard Deviation
Overall	249	131	825	386	400.0	124.7
Male	124	131	761	406	404.7	125.7
Female	125	136	825	373	395.4	124.0
Second grade, elementary						
Both	60	136	736	445	448.0 *	133.6
Male	28	248	718	438	450.0	127.8
Female	32	135.6	735.8	467	446.4	140.5
Fifth grade, elementary						
Both	53	131	592	346	370.7	110.7
Male	27	131	586	385	393.9	124.2
Female	26	156	592	337	346.6	91.0
First grade, junior high						
Both	56	166	761	383	390.0	109.1
Male	30	166	761	395	393.2	117.7
Female	26	242	577	371	386.3	100.6
Second grade, junior high						
Both	80	218	825	353	390.6	129.0
Male	39	218	723	353	388.7	128.5
Female	41	223	825	353	392.4	131.2

* Significantly higher than fifth grade elementary (*p* = 0.006) and second grade junior high (*p* = 0.039), significantly higher than fifth grade elementary (*p* = 0.013) (one-way ANOVA, Bonferroni’s test).

**Table 5 antioxidants-11-01570-t005:** Veggie Meter scores of 249 students at three measurement points.

	Baseline	Month 3	Month 6
Min–max	131–825	145–903	151–870
Median	386	413	428
Mean	400.0	436.0 *	447.4 *
Standard deviation	124.7	136.3	140.4

* Significantly higher than the baseline (*p* < 0.0001) (one-way ANOVA, Bonferroni’s test).

**Table 6 antioxidants-11-01570-t006:** Significant variables in the multivariate logistic regression analysis with model selection for factors exceeding the average Veggie Meter score.

	Regression Coefficient	*p*	Odds Ratio	95% Confidence Interval
Grade	−0.28	<0.0001	0.76	0.66, 0.87
Eating three cups or more of green and yellow vegetables daily	**1.06**	**0.0029**	**2.89**	**1.44, 5.82**
Eating fruit in the last 2 weeks	**1.26**	**0.0386**	**3.54**	**1.07, 11.72**
Drinking vegetable or tomato juice three times or more a week	**1.99**	**0.0019**	**7.30**	**2.08, 25.56**
Eating *Citrus unshiu* in the last 2 weeks	0.50	0.1361	1.64	0.86, 3.16
Improving the dietary habits of the child by parents	0.53	0.1064	1.70	0.89, 3.25

## Data Availability

The data presented in this study are available on request from the corresponding author. The data are not publicly available due to the contract between Hamamatsu City and investigators.

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
