# Peer review of "Improving Skin Carotenoid Levels in Young Students through Brief Dietary Education Using the Veggie Meter"

_antioxidants, 2022, doi:10.3390/antiox11081570_

Round 1

Reviewer 1 Report

The authors investigated the effects of veggie meter intervention on fruit and vegetable intake habits in 261  elementary and junior high students. They found the guidance of the veggie meter helps increase the skin carotenoids by consuming more fruit and vegetable. This work was well designed and the results were clearly present. It can be accepted for publication after minor revision.

1.  In line 47, please add citation for "humans cannot synthesize carotenoids" if possible.

2. In line 54, please cite " Li et al. Arch. Biochem. Biophys. 2010, 504(1): 56-60.

3. In Figure 16,  please correct the typo to "seaweed and Chinese".

Author Response

Thank you for your intimate review. We appreciate reviewers’ useful comments and revised our manuscript accordingly. Our point-by-point responses to the comments follow in the order they were raised. The revised parts are written in red.

Reviewer 1

The authors investigated the effects of veggie meter intervention on fruit and vegetable intake habits in 261  elementary and junior high students. They found the guidance of the veggie meter helps increase the skin carotenoids by consuming more fruit and vegetable. This work was well designed and the results were clearly present. It can be accepted for publication after minor revision.

Thank you for your very positive comments. We revised the manuscript accordingly.

  1. In line 47, please add citation for "humans cannot synthesize carotenoids" if possible.

We cited manuscript by Billy R. Hammond Jr et al.

  1. In line 54, please cite " Li et al.  Biochem. Biophys. 2010, 504(1): 56-60.

Thank you for your notification of a very interesting manuscript. We added it as a reference.

  1. In Figure 16,  please correct the typo to "seaweed and Chinese".

Thank you for your comments. We have studied the difference between seaweed and seaweed Chinese. We think that seaweed is more suitable in our case, because Japanese eat various kinds of seaweed other than seaweed Chinese.

Reviewer 2 Report

It is an interesting study, but I believe that only on the basis of the analyzes performed on the subjects taken in the study, a relevant result cannot be obtained. I believe that paraclinical analyzes are also necessary for a credible result.

Author Response

Thank you for your intimate review. We appreciate reviewers’ useful comments and revised our manuscript accordingly. Our point-by-point responses to the comments follow in the order they were raised. The revised parts are shown in track changes.

Reviewer 2

It is an interesting study, but I believe that only on the basis of the analyzes performed on the subjects taken in the study, a relevant result cannot be obtained. I believe that paraclinical analyzes are also necessary for a credible result.

Thank you for your comments. In the current study, elementary and junior high school students received brief educational session and skin carotenoid evaluation by Veggie Meter, and the change in skin carotenoid levels were examined after 6 months. As a result, the Veggie Meter scores significantly increased from 400.0 ± 124.7 (standard deviation) to 447.4 ± 140.4 at Month 6. Moreover, the percentage of in-crease at Month 6 was negatively correlated with the baseline values. We believe this finding suggests that subjects who became aware of their inferiority tended to make a significant effort to change their behavior.

Nonetheless, we agree with this comment from the Reviewer 2 that a further confirmatory study may be needed to generalize the current results in ordinary society. As we discussed in this manuscript, one of the possible reasons for the present successful increase in Veggie Meter scores was the characteristics of the present subjects. Most of the present subjects belonged to relatively wealthy families, and their parents were thought to be concerned about their children’s education and health so much. Therefore, students and parents well understood the value of vegetables and changed their behavior. We are now expanding our study to students in public schools. We think that we will be able to evaluate the efficacy of dietary intervention using Veggie Meter in subjects that reflect standard of modern Japanese society.

Reviewer 3 Report

In the research work entitled: “Improving skin carotenoid levels in young students through 3 brief dietary education using the Veggie Meter”, the authors state that the novelty of the work depends on: “It is the first report to show a gradual increase in skin carotenoid levels in children using the Veggie Meter intervention”. Unfortunately, other works already tackle the problem in a similar way:

1.     May, K., Jilcott Pitts, S., Stage, V. C., Kelley, C. J., Burkholder, S., Fang, X., ... & Lazorick, S. (2020). Use of the Veggie Meter® as a tool to objectively approximate fruit and vegetable intake among youth for evaluation of preschool and schoolbased interventions. Journal of Human Nutrition and Dietetics33(6), 869-875.

2.     Di Noia, J., & Gellermann, W. (2021). Use of the spectroscopy-based Veggie Meter® to objectively assess fruit and vegetable intake in low-income adults. Nutrients13(7), 2270.

3.     Ermakov, I. V., Whigham, L. D., Redelfs, A. H., Jahns, L., Stookey, J., Bernstein, P. S., & Gellermann, W. (2016). Skin Carotenoids as Biomarker for Vegetable and Fruit Intake: Validation of the ReflectionSpectroscopy Based “Veggie Meter”. The FASEB Journal30, 409-3.

4.     Martinelli, S., Acciai, F., Tasevska, N., & Ohri-Vachaspati, P. (2021). Using the Veggie Meter in elementary schools to objectively measure fruit and vegetable intake: a pilot study. Methods and Protocols4(2), 33.

5.     …anf so on.

I, therefore, believe that this article does not present ideas of novelty such as allowing publication by a prestigious magazine such as Antioxidants.

Author Response

Thank you for your intimate review. We appreciate your useful comments and revised our manuscript accordingly. Our point-by-point responses to the comments follow in the order they were raised. The revised parts are shown in track changes.

Reviewer 3

In the research work entitled: “Improving skin carotenoid levels in young students through 3 brief dietary education using the Veggie Meter”, the authors state that the novelty of the work depends on: “It is the first report to show a gradual increase in skin carotenoid levels in children using the Veggie Meter intervention”. Unfortunately, other works already tackle the problem in a similar way:

Thank you for your comments. There are many articles that show the usefulness of Veggie Meter, as pointed by this reviewer, however we respectfully disagree that there is no novelty in the current study. Previous articles investigated the usefulness of Veggie Meter can be roughly classified into three categories. First one is validation studies that prove the accuracy of Veggie Meter. The second one is studies to investigate factors that associate with Veggie scores. The third one is interventional studies to examine the behavior change by using Veggie meter. We read articles you cited carefully again and summarized them as follows. We think that all of these articles belong to the second category. In these manuscripts, factors that associated with Veggie score were investigated, but their purposes were not to evaluate behavior changes. The number of intervention studies used Veggie Meter in children is quite limited indeed. We referred them in the present manuscript and made a relevant discussion. As described in discussion (line 312-317), Jones et al [32, 34 in the revised version] reported an increase in Veggie scores between the baseline and second measurements but did not show a continuous increase at the third measurement, but other articles didn’t show serial increase of Veggie scores. The manuscript by Jones et al was the first report that showed the increase of Veggie scores between the first and second measurements. Our results showed the increase of Veggie scores in three consecutive measurements in children for the first time. According to your criticism, we revised the sentence “this is the first report to show a gradual increase in VM scores in children using the VM intervention” to “We could show a gradual increase of VM scores in three consecutive measurements in children using the VM intervention.” (line 317, 318) The statement in conclusion was also revised. (line 410,411)

  1. May, K., Jilcott Pitts, S., Stage, V. C., Kelley, C. J., Burkholder, S., Fang, X., ... & Lazorick, S. (2020). Use of the Veggie Meter® as a tool to objectively approximate fruit and vegetable intake among youth for evaluation of preschool and schoolbased interventions. Journal of Human Nutrition and Dietetics, 33(6), 869-875.

We cited this article as a reference 31 (33 in the revised version) in the present manuscript. This observational study investigated factors that associated with Veggie scores in children. They revealed that men and younger age associated with higher scores but no statistically significant differences across racial groups and weight categories. Since this is not an intervention study, the change in Veggie scores was not investigated.

  1. Di Noia, J., & Gellermann, W. (2021). Use of the spectroscopy-based Veggie Meter® to objectively assess fruit and vegetable intake in low-income adults. Nutrients, 13(7), 2270.

This study investigated factors that associated with Veggie scores in low-income adults served by the Special Supplemental Nutrition Program for Women, Infants, and Children (WIC). They revealed nativity, BMI, and fruit and vegetable intake significantly associated with Veggie scores. They also verified test-retest repeatability. But time dependent change in Veggie score was not investigated in this study.

  1. Ermakov, I. V., Whigham, L. D., Redelfs, A. H., Jahns, L., Stookey, J., Bernstein, P. S., & Gellermann, W. (2016). Skin Carotenoids as Biomarker for Vegetable and Fruit Intake: Validation of the ReflectionSpectroscopy Based “Veggie Meter”. The FASEB Journal, 30, 409-3.

The authors validated Veggie Meter (reflection spectroscopy) by comparing resonance Raman spectroscopy to measure skin carotenoid levels and confirmed the usefulness of the Veggie Meter. Since this is not an interventional study, no data on the changes in Veggie score in children following dietary education intervention was shown. We cited similar articles of the same authors in this manuscript to present the accuracy and usefulness of Veggie Meter.

  1. Martinelli, S., Acciai, F., Tasevska, N., & Ohri-Vachaspati, P. (2021). Using the Veggie Meter in elementary schools to objectively measure fruit and vegetable intake: a pilot study. Methods and Protocols, 4(2), 33.

We cited this article as a reference 30 (32 in the revised version) in the present manuscript. This observational study investigated factors that associated with Veggie scores in children. They revealed that Veggie scores and income-status and child age were inversely related. They found no significant association between Veggie scores and age, sex, and race. Since this is not an intervention study, the change in Veggie scores was not investigated.

  1. …anf so on.

I, therefore, believe that this article does not present ideas of novelty such as allowing publication by a prestigious magazine such as Antioxidants.

As detailed above, this study confirmed that the educational approach involving awareness of V/F shortage by quantification using the Veggie Meter was an effective method to change vegetable/fruit taking behavior and increase carotenoid levels in three consecutive measurements in children for the first time. Thus, we believe the novelty of this study deserves to be published in a high standard journal such as Antioxidants.

Round 2

Reviewer 3 Report

The work is easy to read, the results discussed correctly and the conclusions adequate. In my opinion, line 250-251 should be moved to material and method section at line 161.

Author Response

Thank you for your very positive evaluation. We revised our manuscript according to your comments. The sentence "The percentage of increase of VM scores at Month 6 was calculated as follows: (VM score at Month 6 − VM score at baseline)/VM score at baseline." was moved to the section of "2.3. Measurement of skin carotenoid levels".